# The Security and Driving Factors of the Innovative Ecosystem: Evidence from the Yellow River Basin

**DOI:** 10.3390/ijerph20032482

**Published:** 2023-01-30

**Authors:** Yanxia Wu, Shuaishuai Yang, Fangnan Liu, Yuxuan Cao, Mingyue Jiang

**Affiliations:** College of Economics and Management, Xi’an University of Technology, Xi’an 710054, China

**Keywords:** innovative ecosystem, Lotka–Volterra symbiotic coupling model, Granger causality test, the Yellow River Basin

## Abstract

Ecological protection and high-quality development of the Yellow River Basin have always been seriously restricted by its unreasonable economic structure and low level of innovation. A two-dimensional indicator system was constructed to evaluate the security of the innovative ecosystem in the Yellow River Basin and identify its driving factors. The R clustering, the method of coefficient of variation, and the entropy method were used to screen and empower the indicators, and then the Lotka–Volterra symbiotic coupling model was introduced innovatively to calculate the security index and analyze its spatio-temporal evolution. Finally, the fixed-effect regression model and Granger causality test were used to identify the driving factors. It was found that (1) The security of the innovative ecosystem in the Yellow River Basin from 2012 to 2021 showed an overall upward tendency, but it was still at a low level, and the innovative landscape community lagged behind the innovative biome for a long time; (2) The security status of the innovative ecosystem showed a differential pattern of “high in the east and south, and lower in the west and north”; (3) Innovation transformation ability, innovation consumption capacity, education, and natural ecological environment are crucial driving factors for improving the security level of an innovative ecosystem.

## 1. Introduction

In recent years, science and technology innovation for economic development has been highly valued by countries around the world as a new driving force [1,2]. The Chinese government proposed the *Innovation-driving development strategy* in 2012, which emphasizes that, as China’s economy enters a new stage of development, scientific and technological innovation must be placed at the core of overall national development and considered an important driving force for economic development to achieve a comprehensive upgrade of the economic structure [3].

Since Schumpeter proposed the concept of innovation at the beginning of the last century [4], scholars from all over the world have conducted extensive and in-depth research on it. The perspective has been changed from product or technology innovation (linear innovation paradigm) concentrated within a single company to marketing and strategic management innovation, and then to open innovation, platform innovation, innovation system, and innovative ecosystem that are now hotly discussed [4,5]. As the latest paradigm of innovation, the innovative ecosystem has become the research hotspot for its diversity, self-organization, adaptability, strong resilience, and sustainability.

Due to historical, natural, and other reasons, the economic development level of the Yellow River Basin has lagged behind the nation for a long time, as well as the development of science and technology [6]. Additionally, the perennial dependence on unrestricted exploitation of natural resources and the large-scale destruction of the ecological environment have significantly reduced the ecological carrying capacity and resilience, causing greater pressure on economic development. Therefore, the Chinese government has proclaimed “environment protection and high-quality development of the Yellow River Basin” as a major national strategy with great foresight [7]. To successfully realize that goal, it is sensible for us to consider how to improve the scientific and technological innovation capacity of the nine provinces (regions) in the Yellow River Basin.

As a crucial ecological barrier in western China and also an important economic node of the “Belt and Road” strategy, how can the Yellow River Basin coordinate the innovation activities of various innovation actors, improve the efficiency of collaborative innovation, and enhance the contribution of innovation in economic growth with the pre-condition of environmental protection? We think building a scientific, rational, green, and safe innovative ecosystem is an effective response.

Therefore, this paper takes the Yellow River Basin as the empirical object and conducts several works. Firstly, we use the R clustering method, coefficient of variation method, and entropy value method to screen and empower indicators. Secondly, we calculate the security index using the Lotka–Volterra symbiotic coupling model and then analyze the spatial evolution pattern and aggregation by ArcGIS. Finally, the key factors affecting the security of the innovative ecosystem were identified through panel data regression and the Granger causality test, and corresponding enhancement strategies are proposed.

## 2. Literature Review

The innovative ecosystem is one of the typical results of the interdisciplinary application of ecosystems from ecology to socio-economic sciences. Moore was the first scientist who introduced the ecosystem into the business field, arguing that it was too one-sided to consider the development of a particular enterprise alone and that the role of the enterprise in business development must be defined from a broader conceptual level [1]. On this basis, Iansiti and Levien clearly defined the business ecosystem, that is, a multidimensional network of suppliers, distributors, and companies [2], which led to extensive thinking on the extension of the ecosystem. Ander and Kapoor [8,9] extended the concept of the business ecosystem to the field of innovation, insisting that an enterprise’s innovation strategy must match the innovative ecosystem in which it operates. Since then, the innovative ecosystem has gone through three paradigms: linear innovation paradigm based on neoclassical economic theory [10], systematic innovation paradigm based on open innovation theory [11], and ecosystem innovation paradigm based on natural ecosystem theory [12,13,14].

Recently, Chen Jin and Qu Guannan [15] proposed that the innovation paradigm has experienced the following paths according to different dimensions: (1) Attention to market and technology; (2) Attention to the intrinsic meaning and perceived value of the product; (3) Attention to externalities and social needs. From this view, the *Schumpeter-style* innovation of “reconstructing products function and reorganizing factors of production” [16] has been unable to effectively cope with the disruptive impact of many “black crane” and “gray rhino” events on enterprise organizations in modern society, thus giving birth to a new paradigm called *Meaningful Innovation*. It emphasizes that innovation should not only focus on the improvement of product or service performance but also on users’ experience. The studies of the innovation paradigm are shown in Table 1.

Among all of them, the innovative ecosystem has become the hottest topic of current research due to its richer connotation, greater resilience, and more sustainable economic and environmental benefits. In terms of chronological order, research on it initially focused on the basic aspects such as concept and connotation [17], structure and function [18,19], and value-creating and sharing [20]. As research progresses, people began to focus on the measurement of the level of synergy of the innovative ecosystem [21,22] and its spatio-temporal evolution [23,24,25]. In addition, scholars have gradually increased their research in the Chinese scenario and produced many concepts with contemporary characteristics [26].

However, the research content is focused on macro-levels such as regional, industrial, and national levels, and the front-level mechanisms of system operation, few studies concentrate on the security of the innovative ecosystem. Referring to the existing research on security [27,28], we define the security of an innovative ecosystem: it refers to the degree to which innovative biome and innovative landscape community in a certain region influence, interact, and improve each other, and thus promote the evolution of the system to advanced forms, which ultimately contribute to local socio-economic development and ecological environmental protection.

This study may provide new ideas for research in the field of innovation, and further enrich the research perspectives and methods of innovation management. The consideration of security can guide practitioners to continuously improve the construction of a regional innovative ecosystem. The Granger causality test is more economically explanatory, so the driving factors explored by it are also more instructive for economic development and environmental protection.

## 3. Methods and Models

This paper aims to conjecture the security of the innovative ecosystem and identify its driving factors. To achieve these goals, we used the Lotka–Volterra symbiotic coupling model to calculate the comprehensive security index, analyze its spatial and temporal evolution pattern, and finally, we identified the driving factors through fixed-effect panel regression models. The research route of technology is shown in Figure 1.

### 3.1. Overview of the Study Area and Data Sources

The Yellow River Basin is located between 96~119° E and 32~42° N from the Qinghai-Tibet Plateau, covering a total of nine provinces (regions) from west to east, including Qinghai, Sichuan, Gansu, Ningxia, Inner Mongolia, Shaanxi, Shanxi, Henan, and Shandong, which is a biological corridor linking the Qinghai–Tibet Plateau, the Loess Plateau, and the North China Plain. It plays an important role in China’s energy use, chemical production, and raw material supply, and it has a very important strategic position in terms of economic and social development and ecological security. In 2021 the basin’s gross domestic product reaches 286,851.67-billion-yuan (The value was calculated by the GDP of the nine provinces (regions) which was searched in their Statistical Yearbook of 2022), accounting for 25.08% of the whole country, and the total regional population is 421.2 million, accounting for 29.84%. The geographical location in China is shown in Figure 2.

In this paper, panel data of nine provinces (regions) in the Yellow River Basin from 2012 to 2021 are selected for the study, and the data are chosen from nine provincial (regional) statistical yearbooks, national economic and social operation statistical bulletins, China Urban Statistical Yearbook, China Environmental Statistical Yearbook, and China Financial Statistical Yearbook (All the related data can be obtained from the website of the National Bureau of Statistics official). Some of the missing data were completed using interpolation and linear regression.

### 3.2. Construction of the Lotka–Volterra Symbiotic Coupling Model

#### 3.2.1. The Lotka–Volterra Symbiosis Model

The Lotka–Volterra model was first proposed by A. J. Lotka and V. Volterra to quantitatively describe the competition and cooperation relationship among different populations in natural ecosystems [29] (assuming there are two populations in the ecosystem *S*_1_ and *S*_2_):(1)$$\frac{dN_{1}\left(t\right)}{dt}=r_{1}N_{1}\left(t\right)\frac{K_{1}-N_{1}\left(t\right)-\alpha N_{2}\left(t\right)}{K_{1}}$$
(2)$$\frac{dN_{2}\left(t\right)}{dt}=r_{2}N_{2}\left(t\right)\frac{K_{2}-N_{2}\left(t\right)-\beta N_{1}\left(t\right)}{K_{2}}$$
where, *r* is the growth rate of populations; *N* is the existing population number; *K* is the environmental capacity of the ecosystem; *α* (*β*) represents the coefficient of the competitive strength of *S*_2_ to *S*_1_ (or *S*_1_ to *S*_2_); *t* is time series; *αN*_2_ (*t*) represents one unit of *S*_2_ encroachment on *α* unit(s) of *S*_1_ living space; *βN*_1_ (*t*) is the opposite.

#### 3.2.2. The Lotka–Volterra Symbiotic Coupling Model of Innovative Ecosystem

A regional innovative ecosystem can be regarded as complex aggregates composed of innovative biomes and innovative landscape communities. The former includes subjects that organize and participate in innovation activities, namely innovation producers, innovation intermediaries, and innovation consumers. They complete the research and development, transformation, and application of new technologies, which is the main force to maintain the operation of the system. While the latter includes economy, society, education, public facilities, and other innovative elements required to complete innovation activities.

There is a complex coupling interaction mechanism between the two communities. On the one hand, the biomes consume a lot of resources in the process of innovation activities, causing certain damage to the landscape community. But the performance generated by the biomes, in turn, improves and repairs the landscape community. On the other hand, the quality of innovative landscape construction can play a regulatory role in the aggregation and growth of innovative subjects which then affects the development of biological communities [30,31]. It can be deduced that the relationship between the two is consistent with the theory of symbiosis of biological populations in ecology.

Based on the above analysis, this paper carries out the economic translation of the Lotka–Volterra symbiosis model and constructs a regional innovative ecosystem symbiotic coupling model combined with the coupling coordination model.

(1)The translation of the Lotka–Volterra symbiosis model(3)$$\frac{dB\left(t\right)}{dt}=r_{B}B\left(t\right)\frac{D-B\left(t\right)-\alpha E\left(t\right)}{D}$$(4)$$\;\;\frac{dE\left(t\right)}{dt}=r_{E}E\left(t\right)\frac{D-E\left(t\right)-\beta B\left(t\right)}{D}$$
where *r* is the growth rate; *B*(*E*) is the innovative biomes (landscape) community level; *D* is the regional environmental capacity; *α*(*β*) is the coefficient of the competitive strength; *t* is time series; *α* > 0 means that the innovative biomes subsystem is infringed by the innovative landscape subsystem; *α* < 0 means the innovative landscape subsystem expands the survival space of the innovative biomes community subsystem. *β* is the same.

The model is discretized by choosing *k* as the time series variable: (assuming the environmental capacity and coefficient of the competitive strength near year *k* are constant).
$$\alpha \left(k\right)=\frac{\left[\varphi_{B}\left(k\right)D\left(k\right)-B\left(k\right)\right]}{E\left(k\right)}\;$$
(5)$$\beta \left(k\right)=\frac{\left[\varphi_{E}\left(k\right)D\left(k\right)-E\left(k\right)\right]}{B\left(k\right)}$$
in which,
$$\varphi_{B}=1-\frac{B\left(k+1\right)-B\left(k\right)}{B\left(k\right)}\times =1-\frac{\gamma_{B}(k\;+1)}{\gamma_{B}(k)}$$
$$\varphi_{E}=1-\frac{E\left(k+1\right)-E\left(k\right)}{E\left(k\right)}\times =1-\frac{\gamma_{E}(k\;+1)}{\gamma_{E}(k)}$$
where *γ_B_* (*k*) and *γ_E_* (*k*) represent the year-over-year growth rates of *B* and *E*.

The respective security index of the biome subsystem and the landscape subsystem can be constructed based on the meaning of the competition coefficients *α*(*k*) and *β*(*k*).
(6)$$LB=-\alpha \left(k\right)\;$$
(7)$$LE=-\beta \left(k\right)\;$$
where *α*(*k*) and *β*(*k*) represent the competitive strength between biome subsystem and the landscape subsystem; *LB*(*k*) and *LE*(*k*), respectively, represent the security of two subsystems.

To move forward a single step, we build the innovative ecosystem security index:(8)$$L\left(k\right)=\frac{LB\left(k\right)+LE\left(k\right)}{\sqrt{LB^{2}\left(k\right)+LE^{2}\left(k\right)}}$$

(2)The coupling coordination model

This paper uses the coupling coordination model to calculate *D*(*k*) in Equation (5):$$C=\sqrt[{n}]{\frac{U_{1}\times U_{2}\times \cdots \times U_{n}}{{\left(\frac{U_{1}\;+\;U_{2}\;+\;\cdots \;+\;U_{n}}{n}\right)}^{n}}}$$
$$T=\alpha_{1}U_{1}+\alpha_{2}U_{2}+\cdots \alpha_{n}U_{n}$$
(9)$$D=\sqrt{C\times T}$$
where, *T* denotes the comprehensive evaluation value, which describes the overall development level of each subsystem; *D* represents the regional innovative environment capacity; *α_i_* (*i* = *1*, *2*, *3*, …, *n*) is the weight coefficient of each subsystem indicator, which is calculated by the entropy value method.

Then, Equation (8) is the final form of the Lotka–Volterra symbiotic coupling model this paper constructed. *L*(*k*) characterizes the degree of superiority or inferiority of the symbiotic relationship between the biome subsystem and the landscape community subsystem, that is, the security index of the innovative ecosystem. From the mathematical relationship, its value range is [−√2, √2], and the larger the security index, the better the development status of the innovative ecosystem, and the higher the security.

Additionally, the adoption of numerical schemes for Lotka–Volter can be described as follows. Firstly, the *B* (*k* + 1), *B*(*k*), *E*(*k* + 1), *E*(*k*) were calculated by Equation (8), and then we can get the year-over-year growth rates of *B* and *E* which are represented by *γ*. Using *γ* we get $\varphi_{B}\left(k\right)$ and $\varphi_{E}\left(k\right)$ which are used to calculated the coefficient of the competitive strength *α*(*β*), and at last L(K) is calculated.

### 3.3. Indicators Screening and Weights Assignment

#### 3.3.1. Indicators Screening

To improve the scientific accuracy of the selection of indicators and increase the amount of information contained in each indicator, 34 indicators are screened by the R clustering method and the coefficient of variation method. The R clustering method aims to classify the indicators containing the same or similar information in the primary indicators into one category to reduce the correlation. While the coefficient of variation method is designed to calculate the amount of information for the selected indicators, retaining the larger amount of information in each type of indicator.

In this paper, the original data are first standardized, and then used to cluster the indicators and perform nonparametric K–W tests, and finally, the coefficient of variation method (CV value is at 0.5 as the threshold) is used to screen the indicators. According to the above steps, the indicators are screened based on the data of the Yellow River Basin from 2012 to 2021, and the results available in Table 2.

#### 3.3.2. Indicator Weights Assignment

This paper uses the entropy weighting method (for a certain indicator, the entropy value can be used to determine the dispersion degree of a certain indicator. The smaller the entropy value, the greater the dispersion degree of the indicator, and the greater the influence (i.e., weight) of the indicator on the comprehensive evaluation) to assign weights to the 21 indicators screened out, and the results are shown in Table 3.

## 4. Results and Analysis

### 4.1. Spatio-Temporal Evolution Analysis

#### 4.1.1. Time Series Analysis

Through the empirical analysis of the security status of innovative ecosystems in the Yellow River Basin, the comprehensive security index of the nine provinces (regions) is finally obtained, as shown in Figure 3.

As can be seen from Figure 3, the security index in the Yellow River Basin has generally shown an upward trend, among which the index in the Inner Mongolia Autonomous Region has increased the most (from −1.338 in 2013 to 0.788 in 2020), and Shandong Province has the smallest increase (from 1.193 to 1.413).

The reasons can be summarized in two points. On the one hand, as China enters a new stage of development, the economic growth model is changing from the traditional “resource-dependent” to “technology-oriented”, backward production capacity is constantly eliminated, and the economic structure is optimized. In this process, the overall innovation level has been greatly improved, which is especially significant in resource-dependent provinces, such as Shanxi Province. As a traditional coal province, excessive consumption of resources in the past has caused serious damage to its ecological environment [32]. But in recent years Shanxi Province has actively implemented an innovation-driving strategy, and the security level of the innovative ecosystem has increased from −0.541 to 0.889. On the other hand, the ecological protection and high-quality development of the Yellow River Basin is being elevated to a major national strategy [33], the government actively responds to the national call, cultivating various innovative subjects and created a conducive atmosphere for innovative landscapes. In addition, they paid great attention to the coordination of innovative biomes and innovative landscape, so that the security of innovative ecosystems in the basin has shown an upward trend.

Another phenomenon that is easy to detect is the security status which can be divided into two categories “high starting point-low growth rate” represented by Shandong and Sichuan and “low starting point-high growth rate” represented by Inner Mongolia and Gansu. It can be seen that the security of the innovative ecosystem in the Yellow River Basin conforms to the principle of diminishing marginal effect and evolutionary momentum theory in economics. It means that the improvement rate and growth space of provinces and regions with a low starting point are greater than those with higher points.

#### 4.1.2. Spatial Evolution

(1)Spatial distribution pattern

To further reveal the geospatial distribution and differences in the security status, we use ArcGis10.2 to visually express and display the security index of each province in the years 2013, 2015, 2017, and 2019.

From Figure 4, Figure 5, Figure 6 and Figure 7, we can conclude easily that the security pattern of the innovative ecosystem in the Yellow River Basin is quite different and has not improved significantly. Shanxi (2015), Ningxia Hui Autonomous Region and Gansu (2017), and Qinghai (2015) even declined, only Shandong (1.41), Sichuan (1.41), and Shaanxi Province (1.30) were at a relatively high level. The overall security shows a pattern of “higher in the east and the south, lower in the west and the north”.

The reason is that Shandong Province is located in the Bohai Sea Port, with a high degree of openness, and thus the level of economic development is far ahead of other provinces. Shandong has many innovation subjects, and its innovation resources are more concentrated, the innovative landscape is relatively perfect so that new ideas, new technologies, and new products can smoothly move from the laboratory to the market. Although Shaanxi and Sichuan are located in the west of China and the degree of openness is relatively low, at the same time the development environment is relatively poor, both of them have a high-security degree. The reason is that there are a large number of universities, scientific research institutions, rich reserves of innovative talent resources, intermediary service institutions, and a relatively complete innovation infrastructure. However, Qinghai Province and Ningxia are located in the hinterland of northwest China, the economic development conditions are relatively harsh, there are fewer innovation subjects, and the construction of innovative landscapes is not perfect enough, coupled with China’s strategic positioning as “ecological protection main functional area”, the safety improvement of innovative ecosystem is not so obvious.

(2)Spatial correlation analysis

To further explore whether there is a spatial correlation among the security of innovative ecosystems in the Yellow River Basin, this paper uses GeoDa [34] to test the global Moran for the index, and the results show that:(1)The global Moran index based on the Yellow River Basin is negative in 2013, 2015, and 2017, and it only turns positive in 2019. So, we can say that there is a negative or only a weak positive spatial correlation, but none of them passed the significance test. However, this doesn’t mean that there is no spatial correlation in the Yellow River Basin because any geographical thing or attribute is correlated with each other in spatial distribution, and there is clustering, random, and regular distribution;(2)In order to explore the real reason of the negative result. The Yellow River Basin is divided into three sub-regions, eastern, central, and western for the global Moran test. The results show that the Moran index is greater than zero and shows high significance, indicating that there is an obvious spatial positive correlation in the sub-regions (Table 4). The reason is that the Yellow River Basin spans three major plates of eastern, central, and western China, and the traditional *0–1* type geographical weight matrix is not applicable.

### 4.2. Identification of Driving Factors

#### 4.2.1. Correlation Study

In this paper, we are going to explore the driving factors of the security of innovative ecosystem in the Yellow River Basin by taking *L*(*k*) as the explained variable and the innovation production capacity, innovation transformation capacity, innovation consumption capacity, etc. as the explanatory variables. The specific variable settings are detailed in Table 5.

(1)Descriptive statistical analysis

Before performing the correlation analysis, descriptive statistical analysis of the panel data is conducted to understand the basic situation of the data (Figure 8).

As can be seen from the above figure, the observed values of each variable (under the premise of large sample) obey or approximately obey normal distribution with good statistical characteristics, and there are no large range of outliers, which satisfies the basic conditions for analysis using the spatial panel regression model. Therefore, the next step of the correlation study is to build the model and to analyze the results.

(2)Model test and selection

To further determine the model used for regression, the F-test, BP test, and Hausman test are performed on the panel data, and the results are shown in Table 6.

From the test results, we can see that the F-test shows a significance of 5% level (F (8,56) = 5.273, *p* = 0.000 < 0.05), indicating that the FE model fits better than the POOL model. The BP-test also shows a significance of 5% level (chi (1) = 3.586, *p* = 0.029 < 0.05), which means that the RE model fits better than the POOL model. Hausman’s test shows a significance of 5% level (chi (7) = 18.653, *p* = 0.009 < 0.05); therefore, the FE model is superior to the RE model. In summary, the regression analysis was finally carried out with the FE model.

Based on the panel data of the provinces in the Yellow River Basin from 2012 to 2021, referring to the research of Yanxia Wu et al. [20,21], we built an econometric model for studying the security dynamics of innovative ecosystems:(10)$$L(k)=\beta_{i}+\beta_{t}+\beta_{1}IPC_{it}+\beta_{2}ITC_{it}+\beta_{3}ICC_{it}+\beta_{4}EDP1_{it}+\beta_{5}SDP_{it}+\beta_{6}EDEP2_{it}+\beta_{7}NDP_{it}\;\;+\varepsilon_{it}$$
where *i* is the region; *t* is the year; *β_i_* is individual effect; *β_t_* is time effect; *ε_it_* is random perturbation term satisfying E=0.

(3)Analysis of regression results

The FE model is used to carry out panel data regression to explore the relevant influencing factors of innovative ecosystem security in the Yellow River Basin, and the results are shown in Table 7.

It can be seen that IPC shows a significance of 5% (*t* = 1.935, *p* = 0.035 < 0.05), and the regression coefficient value is 1.595 > 0, indicating that IPC has a significant influencing effect on L(K). Likewise, ITC shows a significance of 5% (*t* = 2.606, *p* = 0.012 < 0.05) and the regression coefficient value of 2.751 > 0, which means that ITC has an influencing effect on L(K). The same applies to ICC, EDP1, SDP, EDP2, and NDP analyses. Therefore, we can conclude that the relevant factors of the security of the innovative ecosystem in the Yellow River Basin are IPC, ITC, ICC, EDP_2_, and NDP.

It is thus clear that as a complex system integrating the production, transformation, and consumption of new technologies, the ultimate performance of the innovative ecosystem depends not only on the unilateral innovation production capacity, innovation transformation capacity, or innovation consumption capacity, but also on the natural ecological environment and the level of education.

#### 4.2.2. Causality Study

Based on the correlation analysis, the driving factors with the Granger causal relationship to the security of the innovative ecosystem are further determined by the Granger causality test. This can make the conclusions more policy-oriented. To prevent the occurrence of false regression, the unit root test is performed to verify its smoothness, and the result is shown in Table 8.

From the results of the unit root test, we can see that the *p* value of the selected panel data is significant at the level of 1%, rejecting the null hypothesis (H_0_: the sequence is non-stationary series), indicating that the sequence is stationary. The Granger causality test can be performed directly and the test results are shown in Table 9.

From the test results, we can see that for the sample *IPC* and *L*(*K*), the significance *p* value is 0.122, which does not show significance, so we should accept the null hypothesis. That means the change of *IPC* cannot cause an obvious change of *L*(*K*), that is, the improvement of innovation production capacity is not the main reason for the improvement of the security of the innovative ecosystem in the Yellow River Basin. The reason may be in the modern society with a highly developed commodity economy, the improvement of new product production capacity is no longer the primary problem that the innovation ecosystem needs to solve, whereas how to improve the conversion rate of new technologies and the innovation environment is the key to improving the security of the innovative ecosystem.

Similarly, for the sample *ITC* and *L*(*K*), the significance *p* value is 0.001, showing a significance level of 1%, rejecting the null hypothesis. That means the change of *ITC* may cause a significant change in *L*(*K*), that is, the improvement of innovation transformation ability is the main reason for the improvement of the safety of the innovative ecosystem (The same is true for the analysis of other factors).

It should be noted that the “reason” here is not the real “cause and effect” relationship in logic, but the former has a certain “prediction” or “explanation” function for the latter, which is more in line with the economic sense of “cause and effect” and more practical significance.

#### 4.2.3. Driving Factors for the Security of Innovative Ecosystem

Combined with correlation analysis and causality analysis, it can be rigorously concluded that the driving factors of innovative ecosystem security in the Yellow River Basin are innovation transformation capacity (*ITC*), innovation consumption ability (*ICC*), education driving force (*EDP*_2_), and natural driving force (*NDP*).

The level of *ITC* directly determines whether innovative products can be smoothly converted into economic benefits. In the process of innovation transformation, technology incubators, financial institutions, legal consulting institutions, and other intermediaries play an important role, which is a key part of innovative ideas to cross the “Valley of Death” [35] and “Darwinian Sea”. A very important reason why the security of the innovative ecosystem in the Yellow River Basin shows a different distribution pattern of “high in the east and south part and low in the north and the west part” is that the latter lack high-level innovation intermediary service institutions, which results in a low rate of innovation conversion.

*ICC* is an important criterion for measuring the market acceptance of innovation products and services. It is also a key node for the innovative ecosystem to achieve effective feedback, which plays a crucial role in the security of it. On the whole, the innovation consumption capacity in the Yellow River Basin is not so high, and there is still a big gap between the number of high-tech enterprises, per capita disposable income, and the potential of the innovation consumer market compared with the Yangtze River Basin [27]. This is especially obvious in the provinces in the upstream region.

*EDP*_2_ provides high-quality innovation talent resources for the innovative ecosystem, driving the improvement of innovation production capacity and innovation transformation capacity. At the same time, the enhancement of population quality can also improve the innovation consumption potential, thereby making the innovative ecosystem evolve towards a safer and more efficient form.

As an emerging factor in the socio-economic field, *NDP* begins to play a necessary role in the process of improving the safety of the innovative ecosystem. *NDP* provides geographical space for the development of various innovation activities [24], that’s why all of the activities must not damage the natural ecological environment. One of the reasons for the low innovative ecosystem security of Shanxi and Qinghai provinces is that the natural ecological environmental protection pressure is very huge, and there is no more capital, manpower, or other investments for innovation activities.

## 5. Conclusions and Revelations

### 5.1. Conclusions

Ecological protection and high-quality development of the Yellow River Basin is one of the major strategies in the new stage of China’s development. Due to natural conditions and geographical location, the Yellow River Basin has various problems of unreasonable industrial structure, backward production capacity, and low innovation levels, which lead to its low security of innovative ecosystems. In this paper, the safety of innovative ecosystems in nine provinces (regions) of the Yellow River Basin is evaluated. Then the main driving factors are analyzed through spatial econometric testing. With all the studies considered, the following conclusions are drawn:(1)From the perspective of time series, on the whole, the security of the innovative ecosystem in the Yellow River Basin has shown a trend of increasing year by year, and the regional innovation subjects and innovation landscape have been greatly developed. However, in general, security is still at a low level, only some years in Shandong, Sichuan, and Shaanxi provinces have reached a high-security level. In addition, the security status of innovative ecosystems can be divided into two categories: “high starting point-low growth rate” and “low starting point-high growth rate”. To rationally allocate the resources and elements so that the two can be efficiently synergistic and thus promote the innovative ecosystem to shift to high-quality development is a problem that must be considered in the future;(2)From the perspective of spatial pattern, there exists great variations in the security status of innovative ecosystems in the Yellow River Basin, showing a differential pattern of “higher in the east and south, lower in the west and north”. The details are listed as follows. Firstly, Shandong, Shaanxi, and Sichuan Province have high security, and the coordinated development of innovation subjects and landscapes has been well realized. Secondly, the security level of innovative ecosystems in Ningxia Hui Autonomous Region and Qinghai Province has been at a low level and even has been deteriorating, which seriously restricts their development. Thirdly, Gansu Province is in a state of fluctuation and has not yet entered a high level of security. Finally, the security situation in Inner Mongolia Autonomous Region, Shanxi, and Henan is remaining the same or is slightly improving, and it is still necessary to break through bottlenecks and build a safer innovative ecosystem in the future;(3)From the perspective of spatial correlation, there is no obvious spatial correlation and spatial aggregation effect when the Yellow River Basin is seen as a whole, but there is an obvious spatial aggregation effect when it is divided into three sub-regions, indicating that the improvement of the security of the innovative ecosystem in the Yellow River Basin needs to be tailored to local conditions, and different guidelines and policies need to be formulated according to the specific location of different regions, rather than generalization;(4)Through correlation analysis and the Granger causality test, we find that the factors related to the security level of innovative ecosystems include innovative production capacity, innovative transformation ability, innovation consumption ability, education driving force, and natural ecology driving force. The factors that have a Granger causality relationship with the security of the innovative ecosystem include innovation transformation capacity, innovative consumption capacity, education driving power, and natural ecological driving power. The level of these factors will directly lead to the improvement or decrease of the security of the innovative ecosystem, so the provinces should continue to make efforts from the above aspects.

### 5.2. Management Revelations

(1)All provinces and regions in the Yellow River Basin should accelerate the construction of a high-level, sustainable and safe innovative ecosystem. On the one hand, they should continue to increase investment in innovation, and cultivate various innovation subjects. On the other hand, they are supposed to pay attention to the construction and improvement of innovative landscapes with establishing all kinds of technology incubations as well as intermediary institutions. At the same time, it is sensible to ensure the maturity of new technologies and enterprises by constantly improving the system of laws and regulations, creating a good atmosphere for innovation;(2)Improve the frequency and efficiency of spatial linkage in all provinces. The security of the innovative ecosystem has a spatial spillover effect [28]. For example, Shaanxi Province has a high level of technological innovation, giving full play to its spatial spillover effect will produce a positive effect on Gansu and Inner Mongolia. If they can implement a scientific and reasonable overall development plan for the nine provinces (regions) according to local conditions, the security of the overall innovative ecosystem will be promoted to a new height;(3)Provinces in the Yellow River Basin should focus on improving innovation transformation capacity, innovation consumption capacity, education level, and natural ecological level. First of all, they should improve the transformation rate of innovation achievements. Secondly, promoting regional innovative consumption capacity and getting more feedback from the consumer market are the right pathways. What’s more, increasing investment in education, especially higher education, is necessary. Finally, they must dedicate themselves to protecting the natural environment so as to lay a solid foundation for the construction and operation of the innovative ecosystem.

## 6. Contributions and Limitations

### 6.1. Contributions

As the latest paradigm of innovation, the basic issues of innovative ecosystem such as its concept, connotation, structure, and functions have been extensively studied. But more in-depth issues such as its internal operation mechanism and operation effect evaluation are still not fully explored. This paper evaluated the security of the innovative ecosystem and explored its driving factors, which are theoretically innovative and can help promote the innovation level of the Yellow River Basin.

Theoretically speaking, the coupling coordination model and the Lotka–Volterra symbiosis model are combined to construct the Lotka–Volterra symbiotic coupling model to measure the security of the innovative ecosystem. The results show that the model has certain validity for the operational evaluation of complex systems and can be used for the comprehensive evaluation of large-scale socio-economic composite systems (e.g., ecological environmental protection system; ecological urbanization system). In addition, screening the primary selection indicators by the R clustering and coefficient of variation method can greatly improve the scientific nature of the indicator system while reducing the workload in evaluation studies. What’s more, it is feasible to explore the Granger causality relationship between variables through the Granger causality test. Granger causality is different from the true sense of causality, which has a certain predictive effect on the explanatory variables, so it has a strong application value in the study of time series and socio-economic contexts. This paper uses the Granger test to explore the driving factors of innovative ecosystem security in the Yellow River Basin, which is somewhat innovative.

From a practical standpoint, the study can, directly or indirectly, help improve the level of innovation of the Yellow River Basin and also provide indicative recommendations to other regions to some extent. What’s more, the study can also help to optimize the economic structure, transform the momentum of development, and change economic development, which are important for high-quality development of China and even all over the world.

Finally, this study can indirectly protect the ecological environment by improving the innovation ability, optimizing its economic structure, and eliminating highly polluting and energy-consuming industries. In addition, the innovative ecosystem security evaluation index system constructed in this paper includes the landscape community subsystem, which contains the measurement indicators related to the natural environment, so the security calculated by this institute can reflect the environmental level of the research area to a certain extent, and can play a certain role in promoting its environmental protection practice.

### 6.2. Limitations

Although this paper covers many aspects of the selection of indicators, and uses the R clustering method and coefficient of variation method to screen them, which can, to a certain extent, ensure the rationality of the index system. However, the innovative ecosystem involves a sea of subjects, the interaction paths are extremely complex, and there is still no unified and authoritative index system for reference. Thus, future research still needs to strengthen the improvement of the innovative ecosystem security rating index system.

What’s more, this paper selects the panel data of the Yellow River Basin over the past 10 years, and the sample size is relatively small, which may have an impact on the final results. Therefore, future research should expand the sample size to enhance the credibility of the research.

## Figures and Tables

**Figure 1 ijerph-20-02482-f001:**
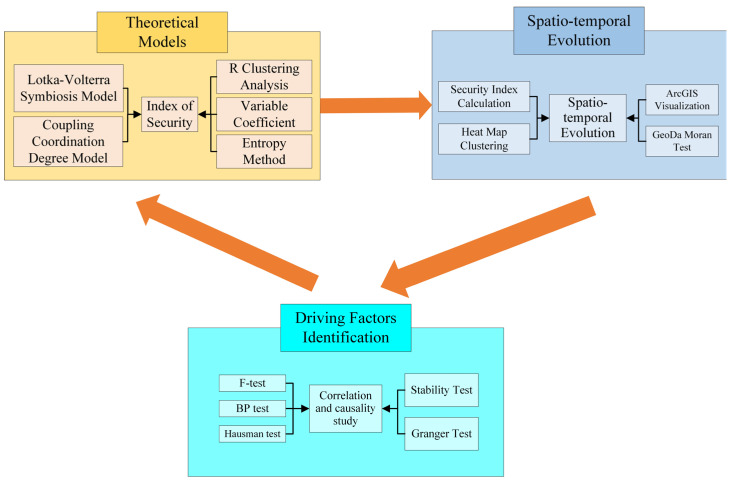
The research route of technology.

**Figure 2 ijerph-20-02482-f002:**
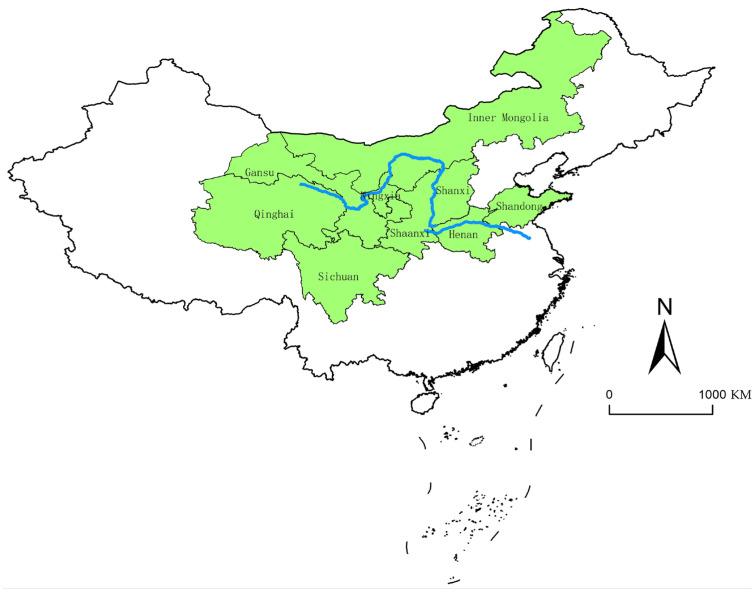
Geographical location of the study area in China.

**Figure 3 ijerph-20-02482-f003:**
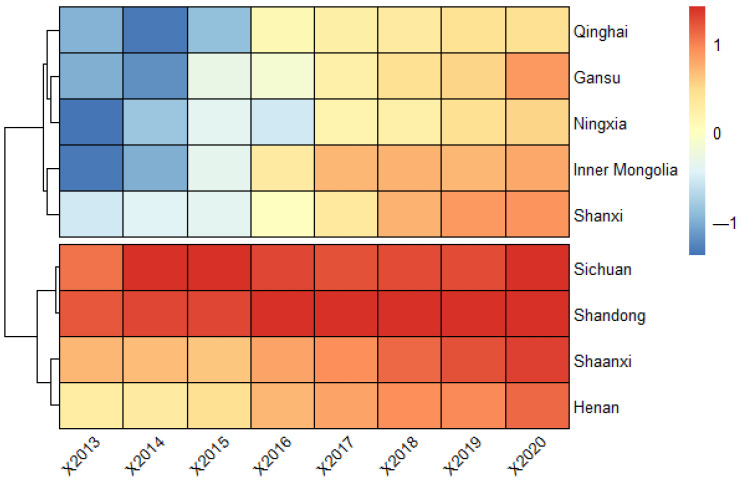
Measurement results of the security index of the Yellow River Basin.

**Figure 4 ijerph-20-02482-f004:**
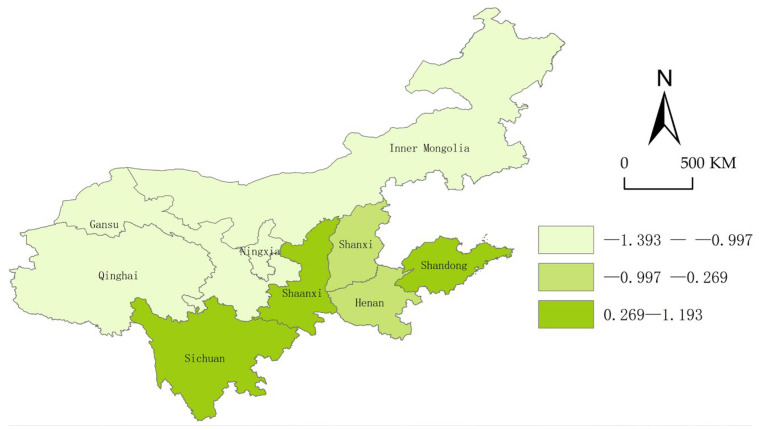
Security landscape of the Yellow River Basin (2013).

**Figure 5 ijerph-20-02482-f005:**
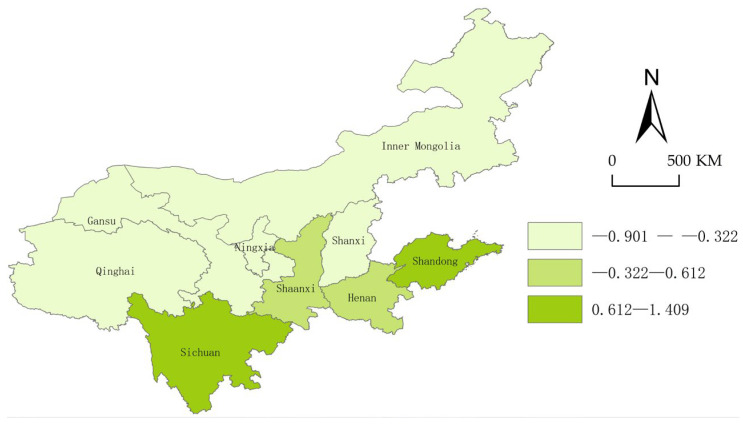
Security landscape of the Yellow River Basin (2015).

**Figure 6 ijerph-20-02482-f006:**
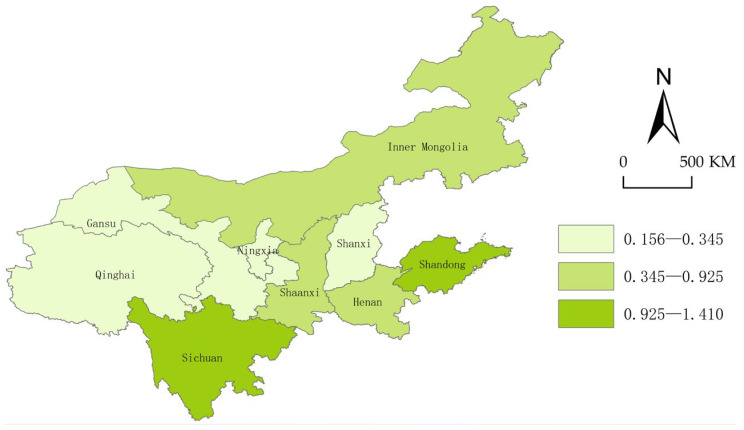
Security landscape of the Yellow River Basin (2017).

**Figure 7 ijerph-20-02482-f007:**
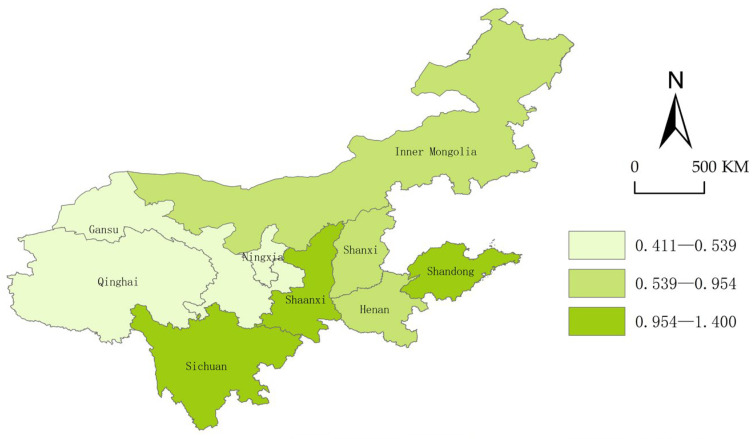
Security landscape of the Yellow River Basin (2019).

**Figure 8 ijerph-20-02482-f008:**
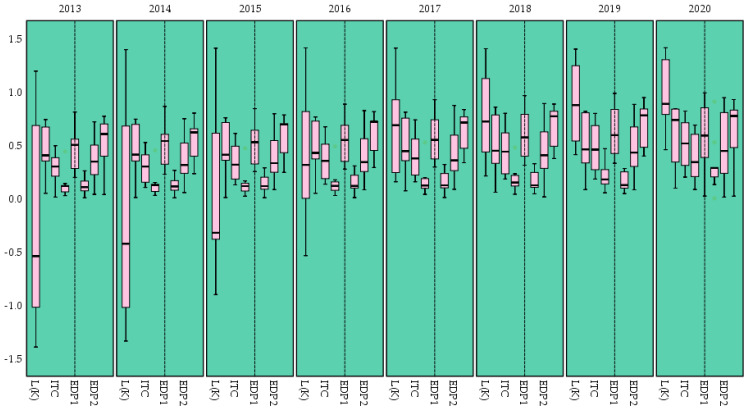
Descriptive statistical analysis results.

**Table 1 ijerph-20-02482-t001:** Evolution of innovation paradigm.

Innovation Paradigm	Theoretical Basis	Core Thoughts
Paradigm 1: Linear Innovation	Neoclassical economic theory	Focus on upgrading product technology for a single enterprise
Paradigm 2: System innovation	Open innovation theory	Focus on the synergy of innovation subjects with industry or sector as the target
Paradigm 3: Ecosystem Innovation	Natural ecosystem theory	Focus on region, emphasizing the synergy between the innovation subject and the environment
Paradigm 4: Meaningful Innovation	Maslow’s theory of needs	Focus on the satisfaction of high-level user experience and the contribution to society

**Table 2 ijerph-20-02482-t002:** Indicator screen results.

Target Layer	Criterion Layer	Indicator Layer	Cluster Results	K-W Test Value	Coefficient of Variation	Filter Results
Innovative Biome	B_1_: Innovation producer	B_11_: Number of universities and research institutions	I	0.897	0.638	Y
		B_12_: Number of general high schools			0.125	N
		B_13_: Number of graduates from higher education institutions			0.569	Y
		B_14_: Number of graduate students in higher education	II		0.236	N
		B_15_: R & D personnel full time equivalent	III		0.568	Y
		B_16_: Annual Patent Grant	IV		0.321	N
	B_2_: Innovation intermediary	B_21_: Number of science and technology incubation bases	I	0.725	0.861	Y
		B_22_: Technology contract turnover as a ratio of GDP	II		0.736	Y
		B_23_: Number of legal institution practitioners	III		0.254	N
		B_24_: Number of employees in urban non-private units in the financial industry			0.698	Y
		B_25_: Persons employed in urban units in transportation, warehousing, and postal services			0.132	N
	B_3_: Innovation consumers	B_31_: Number of high-tech enterprises	I	0.812	0.901	Y
		B_32_: Industrial enterprises above designated size Expenditure as a proportion of GDP	II		0.825	N
		B_33_ ^1^: Market consumption potential of innovative products	III		0.785	Y
		B_34_: Engel coefficient	IV		0.356	N
E: Innovative landscape communities	E_1_: Economical landscape	E_11_: GDP per capita	I	0.925	0.859	Y
		E_12_: Per capita disposable income			0.834	Y
		E_13_: Value added of accommodation and catering	II		0.256	N
		E_14_: Real estate main business income	III		0.315	N
		E_15_: Total retail sales of consumer goods	IV		0.769	Y
	E_2_: Social landscape	E_21_: Number of public libraries	I	0.753	0.754	Y
		E_22_: The number of legal entities in cultural and related industries above the designated size			0.635	Y
		E_23_: Local fiscal public security expenditure	II		0.315	N
		E_24_: Number of Internet broadband access users	III		0.659	Y
		E_25_: Urban road area per capita	IV		0.321	N
		E_26_: Number of buses per 10,000 people			0.598	N
	E_3_: Educational landscape	E_31_: R & D funding for universities and scientific research institutions	I	0.635	0.859	Y
		E_32_: Local fiscal expenditure on education	II		0.785	Y
		E_33_: Local fiscal expenditure on science and technology	III		0.612	Y
	E_4_: Natural landscape	E_41_: Number of parks	I	0.720	0.115	N
		E_42_: Number of national nature reserves	II		0.296	N
		E_43_: Local fiscal environmental protection expenditure	III		0.634	Y
		E_44_: Vegetation coverage	IV		0.495	Y
		E_45_: Green area of built-up area			0.534	Y

^1^ Innovative product market consumption potential = per capita disposable income—per capita living consumption expenditures.

**Table 3 ijerph-20-02482-t003:** Indicator weights.

Target Layer	Criterion Layer	Indicator Layer	Weight
B: Innovative Biome	B_1_: Innovative producer	B_11_: Number of universities and research institutions	0.0374
		B_13_: Number of graduates from higher education institutions	0.0371
		B_15_: R & D personnel full time equivalent	0.0529
	B_2_: Innovation intermediary	B_21_: Number of science and technology incubation bases	0.0475
		B_22_: Technology contract turnover as a ratio of GDP	0.0562
		B_24_: Number of employees in urban non-private units in the financial industry	0.0434
	B_3_: Innovative consumers	B_31_: Number of high-tech enterprises	0.0767
		B_32_: Industrial enterprises above designated size Expenditure as a proportion of GDP	0.0477
		B_33_ ^1^: Market consumption potential of innovative products	0.0137
E: Innovative landscape communities	E_1_: Economical landscape	E_11_: GDP per capita	0.0149
		E_12_: Per capita disposable income	0.0151
		E_15_: Total retail sales of consumer goods	0.0470
	E_2_: Social landscape	E_21_: Number of public libraries	0.0210
		E_22_: The number of legal entities in cultural and related industries above the designated size	0.0691
		E_24_: Number of Internet broadband access users	0.2297
	E_3_: Educational landscape	E_31_: R & D funding for universities and scientific research institutions	0.0644
		E_32_: Local fiscal expenditure on education	0.0261
		E_33_: Local fiscal expenditure on science and technology	0.0427
		E_43_: Local fiscal environmental protection expenditure	0.0252
		E_44_: Vegetation coverage	0.0242
		E_45_: Green area of built-up area	0.0081

^1^ Innovative product market consumption potential = per capita disposable income—per capita living consumption expenditures.

**Table 4 ijerph-20-02482-t004:** Global Moran index and test results.

Year	Global Moran Index	Z-Score	*p*-Value	Partition Moran I Index	Z-Score	*p*-Value
2013	−0.188	−0.218	0.827	0.156	0.128	0.001 ***
2015	−0.145	−0.086	0.932	0.141	0.219	0.000 ***
2017	−0.021	0.360	0.719	0.247	0.411	0.000 ***
2019	0.017	0.506	0.613	0.311	0.536	0.000 ***

*** Represents a significance level of 1%.

**Table 5 ijerph-20-02482-t005:** Security driving factors analysis variable settings.

Variable Type	Variable Name	Symbol	Variable Description
Explained variables	Innovative ecosystem security	L (k)	Characterize the security posture of the innovative ecosystem
Explanatory variables	Innovation production capacity	IPC	Characterize the output capacity of the innovative ecosystem
	Innovation transformation capacity	ITC	Characterize the transformation capacity of the innovative ecosystem
	Innovation consumption capacity	ICC	Characterize the consumption power of the innovative market
	Economic driving power	EDP_1_	Characterize the economic environment of the innovative ecosystem
	Social driving power	SDP	Characterize the socio-environmental status of the innovative ecosystem
	Educational driving power	EDP_2_	Characterize the state of the educational environment of the innovative ecosystem
	Natural driving power	NDP	Characterize the natural ecological environment of the innovative ecosystem

**Table 6 ijerph-20-02482-t006:** Model optimization and selection results.

Type of Test	Purpose of the Test	Test Value	Conclusion
F-test	Selection of FE model and POOL model	*F* (8,56) = 5.273, *p* = 0.000	FE model
BP-test	Selection of RE model and POOL model	χ^2^(1) = 3.586, *p* = 0.029	RE model
Hausman-test	Selection of RE model and FE model	χ^2^(7) = 18.653, *p* = 0.009	FE model

**Table 7 ijerph-20-02482-t007:** Regression results of factors related to the security.

Item	Coef.	Std. Err	*t*	*p*
Intercept	−2.379	0.799	−2.976	0.004 **
IPC	1.595	2.424	1.935	0.035 *
ITC	2.751	1.056	2.606	0.012 *
ICC	2.612	1.342	1.946	0.007 **
EDP_1_	−1.671	2.037	−0.820	0.416
SDP	−1.523	0.826	−1.843	0.071
EDP_2_	−4.174	1.654	−2.523	0.014 *
NDP	5.587	1.183	4.724	0.000 **

Note: F (7,56) = 10.403, *p* = 0.000, R^2^ = 0.191, R^2^(within) = 0.565, * *p* < 0.05, ** *p* < 0.01.

**Table 8 ijerph-20-02482-t008:** Panel data stationariness test results.

The Order of Difference	*t*	*p*	AIC	Critical Value		
				1%	5%	10%
0	−3.991	0.001 ***	99.516	−3.541	−2.909	−2.592
1	−9.375	0.000 ***	108.817	−3.539	−2.909	−2.592

Note: *** represents significance levels of 10%, respectively.

**Table 9 ijerph-20-02482-t009:** Results of the Granger causality test.

Paired Sample		F-Statistic	*p*-Value
IPC	L(K)	7.865	0.122
ITC	L(K)	11.81	0.001 ***
ICC	L(K)	21.103	0.000 ***
EDP_1_	L(K)	2.216	0.141
SDP	L(K)	9.673	0.313
NDP	L(K)	3.378	0.070 *
EDP_2_	L(K)	0.618	0.003 ***

Note: *** and * represents significance levels of 1% and 10%, respectively.

## Data Availability

The data presented in this study are openly available in nine provincial statistical yearbooks, national economic and social operation statistical bulletins, China Urban Statistical Yearbook, China Environmental Statistical Yearbook, and China Financial Statistical Yearbook.

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
