# Peer review of "The Security and Driving Factors of the Innovative Ecosystem: Evidence from the Yellow River Basin"

_ijerph, 2023, doi:10.3390/ijerph20032482_

Round 1
Reviewer 1 Report
Dear authors.
Please find below comments and suggestions regarding your manuscript.
General comment:
Manuscript needs to be reorganized sow that reader can easily go through it and read it. There are missing some critical poits in the manuscript which needs to be addresed. Also manuscript needs to be written according journal author guidelins expecially when citing references and literature review. Also for the review purpose put row numbers in the manuscript. English language needs to be corrected.
Specific remarks:
Abstract is too long and it should be shortened.
Introduction section must be expanded and proper references are needed. Describe in short what is the goal of your research.
In section 2 are missing references [8] and [9]. Also put reference for Lotka Voltaire model and ArcGIS software.
From where are taken results for Table 1? Put reference...
Put reference for l"However, the research content is focused on macro-levels such as regional, industrial, and national levels, and the front-level mechanisms of system operation so far, few studies have focused on the security of the innovation ecosystem."
Put reference for Lotka Voltaire model in section 3.1.1.
Put section 3.3. at the beginning of section 3.
Put appropriate references in section 3.1.2.
Define paprameters in equations (6) and (7).
Section 3.2.1. - define which 34 indicators.
Define entropy weighting method in section 3.2.2.
In table 3 are weights for 20 parameters not 21 as mentioned in text.
Please put references for section 3.3. regrading gross domestic product.
Mark Yellow River and basin on figure 2.
Put references for China Urban Statistical Yearbook, China Environmental Statistical Yearbook, and China Financial Statistical Yearbook.
In last paragraph of section 3.3. put table with data that have been chosen from nine provincial and regional statistical yearbooks, national economic and social operation statistical bulletins...
Put references for section 4.1.1. and 4.2.2. and for other sections...
Use the some colors and legend to define safety index on figures 4, 5, 6 and 7 as same for figure 3 if possible because on figures 4, 5, 6 and 7 last three colours in legend are almost the same (green, yellow).
Section (2) spatial correlation-define GeoDa and Global Moran test and put references.
Why didint you define Yellow River Basin into 9 regions as mentioned before for global Moran test but only 3?
Try to present results in graphical way if it is possible for 9 regions which would better explain and show readres the results you presented in manuscript.
Chapter 6. Please give some guidelines for future work.
Author Response
We have addressed all the comments you mentioned and uploaded it in the attachment, please check it and thank you very much!

Reviewer 2 Report
The authors have drafted a paper that reads pretty well, They present a lot of data information and summaries. However, the authors generally fail to describe what the results mean. What are their significance and utility? See technical review provided.

Author Response

(The authors gave the same response as above.)

Reviewer 3 Report
The major revision is suggested by the reviewer.
1.Because the Lotka-Volterra model has been proposed for almost one hundred years, please describe the novelty and innovation of this manuscript. How does it become superior to other techniques?
2.Please describe the adoption of numerical schemes for Lotka-Volterra model.
3.Please describe how to calibrate and validate Lotka-Volterra model.
4.Please describe how to minimize the calculation error for Lotka-Volterra model.
5.Please discuss the difficulty of model building process.
6. The application of this study is specific and limited because Yellow River is a great and long river with extremely large basin. Please describe the applicability.
7. Please discuss how to apply the results in environmental management.
Author Response

(The authors gave the same response as above.)

Reviewer 4 Report
The authors present an interesting study of innovation at the ecosystem level; however, before the manuscript is published, some parts need to be improved.
(1) The title does not draw the reader's attention, I suggest the authors revise it by making use of the keywords that recall the innovative content of their study.
(2) The abstract is too long and unclear, not providing readers with an effective overview of the research. I advise the authors to follow the journal guidelines: "The abstract should be a total of about 200 words maximum. The abstract should be a single paragraph and should follow the style of structured abstracts, but without headings: 1) Background: Place the question addressed in a broad context and highlight the purpose of the study; 2) Methods: Describe briefly the main methods or treatments applied. Include any relevant preregistration numbers, and species and strains of any animals used. 3) Results: Summarize the article's main findings; and 4) Conclusion: Indicate the main conclusions or interpretations. The abstract should be an objective representation of the article: it must not contain results which are not presented and substantiated in the main text and should not exaggerate the main conclusions.".
(3) The authors refer to the concept of "security of the innovation ecosystem" in the introduction, as already in the title and abstract, but without explaining it. The authors need to provide readers with an adequate background of what it is by referring to the literature.
(4) Still in the introduction, it would be appropriate for the authors to stress the innovative contribution of their study more than the state of the art.
(5) In section 6.1.Contributions, it is necessary for the theoretical and practical implications of the findings of this research to emerge in more generalizable and transferable terms to other socio-economic contexts. Indeed, the Yellow River Basin case study is a means of demonstrating the validity of an analytical approach and not the ultimate goal.
Author Response

(The authors gave the same response as above.)

Round 2
Reviewer 1 Report
Dear authors please find below comments and suggestion for your manuscript.
The manuscript has been revised a lot according to my and other reviewers' comments and instructions, but some parts still need to be revised.
Introduction section is still mising proper references for citing descriptions and claims in it...
Section Results and Analysis needs some more "analsis":
Correct names of "Figures".... i.e. You write "Figure 4", and for Figure 5 you write "Fig 5"... Also some other parts of manuscript needs to be corrected according "Instructions for authors".
Figures 4-5: Please write the names of the provinces on the pictures or their abbreviations to make it easier for readers connect figures and text... This is also suggestion for Figure 2.
Please analyse/discuss Figure 8.... compare results, do analsis so that you understand the basic situation of the data as you write in manuscript.
Author Response
Thank you for another careful review of our article, your comments are critical to the improvement of this article, and we have made detailed revisions and modification instructions for your comments, which are uploaded in the form of attachments, please review them, thank you again!

Reviewer 2 Report
The authors have adequately addressed the reviewer comments.
Reviewer 3 Report
The authors have responded to the comments properly and the manuscript is thus improved substantially. Therefore, it can be acceptable for the publication now.
Author Response
Thank you for your careful review and handling of this article, the perfection of your review comments plays a vital role, and we will continue to polish the article until it fully meets the publication requirements, good luck!
Reviewer 4 Report
In this latest version of the manuscript, the authors have incorporated the reviewers' suggestions, so the article is suitable for publication.
Author Response

(The authors gave the same response as above.)
